# Simplifying Multi-Task Architectures Through Task-Specific Normalization

**Mihai Șuteu**                                                                                    *m.suteu16@imperial.ac.uk*
*Data Science Institute, Imperial College London*

**Ovidiu Șerban**                                                                                   *o.serban@imperial.ac.uk*
*Data Science Institute, Imperial College London*

**Reviewed on OpenReview:** *https://openreview.net/forum?id=QNO893OXrS*

## Abstract

Multi-task learning (MTL) aims to leverage shared knowledge across tasks to improve generalization and parameter efficiency, yet balancing resources and mitigating interference remain open challenges. Architectural solutions often introduce elaborate task-specific modules or routing schemes, increasing complexity and overhead. In this work, we show that normalization layers alone offer a remarkably effective and underexplored mechanism for addressing these challenges. Simply replacing shared normalization with task-specific variants already yields competitive performance, suggesting that elaborate task-specific modules are not always required. Building on this insight, we propose Task-Specific Sigmoid Batch Normalization (TS$\sigma$BN), which combines bounded normalization with discriminative learning rates, enabling soft capacity allocation while fully sharing feature extractors. TS$\sigma$BN improves stability across CNNs and Transformers, matching or exceeding performance on NYUv2, Cityscapes, CelebA, and PascalContext, while remaining highly parameter-efficient. Moreover, its learned gates provide a natural framework for analyzing MTL dynamics, offering interpretable insights into capacity allocation, filter specialization, and task relationships. Our findings suggest that task-specific normalization offers a simple, interpretable, and efficient alternative to more complex MTL architectures. Code is available at *https://github.com/xapharius/TSsNorm*

## 1 Introduction

Multi-task learning (MTL) trains a single model to solve multiple tasks jointly, leveraging shared representations to improve generalization and computational efficiency. Despite many successes, MTL remains difficult to understand and control. Core challenges include task interference, where competing gradients from divergent task requirements disrupt joint training (Zhang et al., 2022); capacity allocation, where shared and task-specific resources must be balanced to avoid dominance (Maziarz et al., 2019; Newell et al., 2019); and task similarity, where the degree of relatedness determines how tasks should interact (Standley et al., 2020). Existing approaches typically address only one of these issues. Optimization-based methods focus on mitigating interference by reweighting losses or modifying gradients (Yu et al., 2020; Navon et al., 2022). Soft-sharing architectures attempt to disentangle capacity by adding task-specific modules on top of a shared backbone, but in doing so often introduce significant design complexity in deciding how modules should interact (Misra et al., 2016; Liu et al., 2019). Neural architecture search methods learn to partition networks based on data-driven estimates of task-relatedness (Guo et al., 2020; Sun et al., 2020).

Despite the availability of foundation models, the challenges MTL faces have not disappeared with improved quality of representations. Whenever parameters are jointly optimized, interference and dominance appear as properties of the joint loss over multiple tasks, regardless of the backbone. Recent transformer-specific

MTL methods (Agiza et al., 2024; Yang et al., 2024) report meaningful gains over hard-sharing baselines on pretrained backbones, confirming the problem is still relevant.

In this work, we argue that normalization layers and in particular batch normalization (BN) (Ioffe, 2015) provide a remarkably effective and underexplored mechanism for addressing these challenges in multi-task learning. Our motivation stems from the following observations:

*Capacity Utilization.* While neural networks are heavily over-parameterized, existing approaches struggle to resolve tasks conflicts (Shi et al., 2023), indicating a failure to utilize available network capacity effectively. *Expressivity and Stability.* Normalization layers have proven to be highly expressive, not only do they stabilize and accelerate training (Santurkar et al., 2018; Bjorck et al., 2018), but also demonstrate remarkable standalone performance when used on random feature extractors (Rosenfeld & Tsotsos, 2019; Frankle et al., 2021), in turn allowing them to be leveraged in the context of fine-tuning (Zhao et al., 2024).

*Structured Sparsity.* BN can learn to ignore unimportant features (Frankle et al., 2021) or be explicitly regularized to produce structured sparsity (Liu et al., 2017; Suteu & Guo, 2022). This can be leveraged for MTL when incompatible tasks cannot fully share features without interference and thus require partitioning. *Parameter Efficiency.* Normalization layers are extremely parameter-efficient, taking up typically less than 0.5% of a model's size. This makes them particularly suitable as lightweight universal adapters for applications where models need to scale to multiple tasks (Rebuffi et al., 2017; Bilen & Vedaldi, 2017).

Lastly, while conditional BN layers have been explored in settings with domain shift (Wallingford et al., 2022; Xie et al., 2023; Chang et al., 2019; Deng et al., 2023), these methods focus on the issue of mismatched normalization statistics and use task-specific BN as a domain-alignment tool. Our focus is different: we study single-domain MTL, where all tasks share the same input distribution and normalization does not become a failure mode. In this setting, we show that task-specific BN can provide a simple way to modulate representations via their affine parameters - turning it from a normalization module into a lightweight mechanism for capacity allocation and interference reduction. The extension of BN as the sole mechanism for modulation and interpretability rather than domain alignment remains largely unexplored.

Motivated by these observations, we propose a minimalist soft-sharing approach for MTL, where feature extractors are fully shared and only normalization layers are task-specific. Unlike prior soft-sharing architectures that add complex modules or routing schemes, our design isolates normalization as the sole mechanism for balancing tasks. Building on $\sigma$BN (Suteu & Guo, 2022), we introduce lightweight task-specific gates that modulate feature usage with negligible overhead, making the approach broadly compatible, easy to implement, and resilient to task imbalance. Beyond performance and efficiency, the learned $\sigma$BN parameters naturally form a task-filter importance matrix, enabling a structured analysis of capacity allocation, filter specialization, and task relationships, thus providing a unified interpretability framework for MTL.

**Contributions:**

- A minimal MTL baseline. We show that simply replacing shared normalization with task-specific BatchNorm (TSBN) already delivers competitive performance out-of-the-box, even without elaborate task-specific modules or routing schemes.

- An extended design combining sigmoid normalization with discriminative learning rates. We introduce TS$\sigma$BN which improves stability and scale across CNNs and transformers. This variant achieves superior performance on nearly all benchmarks while remaining parameter-efficient.

- An interpretable analysis framework. The use of $\sigma$BN further provides a natural lens for analyzing MTL dynamics. By interpreting learned feature importances, we obtain structured insights into capacity allocation, filter specialization, and task relationships.

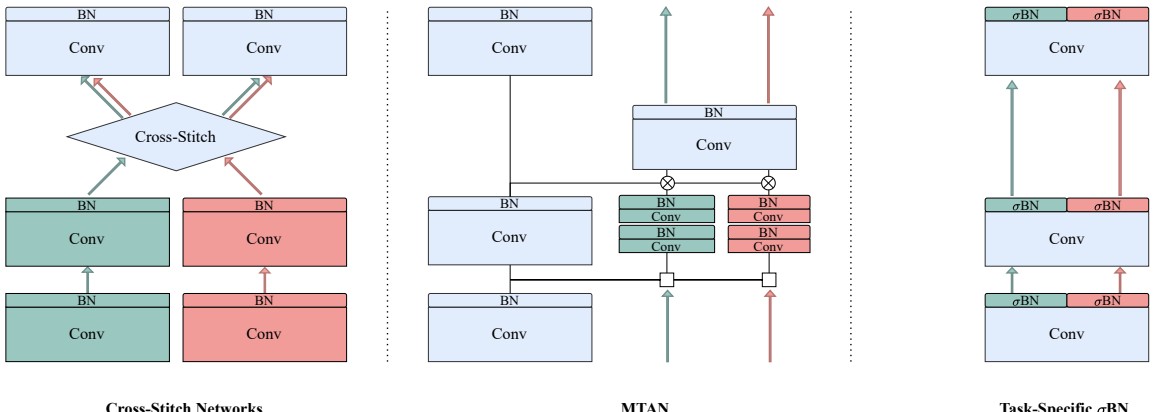

Figure 1: Illustration of soft parameter sharing architectures in a two-task setting. Cross-Stitch Networks (Misra et al., 2016) and MTAN (Liu et al., 2019) incorporate additional feature extractors, which lead to scalability challenges as the number of tasks increases. Task-Specific $\sigma$BN Networks introduce only task-specific normalization layers, offering a highly parameter-efficient solution.

## 2 Related Work

**Soft parameter sharing** methods tackle MTL interference architecturally by introducing task-specific modules to a shared backbone. Design options include replicating backbones (Misra et al., 2016; Ruder et al., 2019), adding attention mechanisms (Liu et al., 2019; Maninis et al., 2019), low-rank adaptation modules (Liu et al., 2022b; Agiza et al., 2024) or allowing cross-talk at a decoder level (Xu et al., 2018; Vandenhende et al., 2020b). However, these methods rely on task-specific feature extractors to avoid negative transfer at the cost of forgoing the multi-task inductive bias. Furthermore, adding task-specific capacity scales poorly with many tasks (Strezoski et al., 2019), and requires extensive code modifications that hinder adaptation to new architectures. Although BatchNorm is present in many of these systems, it is embedded in larger task-specific designs. In contrast, our method isolates BN as the sole soft-sharing mechanism, showing that it is a sufficient solution for competitive MTL while challenging unnecessary complexity.

**Neural Architecture Search (NAS)** methods reduce task interference by choosing which parameters to share among tasks as hard-partitioned sub-networks. Some approaches use probabilistic sampling (Sun et al., 2020; Bragman et al., 2019; Maziarz et al., 2019; Newell et al., 2019) or explicit branching/grouping strategies based on task affinities (Vandenhende et al., 2020a; Guo et al., 2020; Bruggemann et al., 2020; Standley et al., 2020; Fifty et al., 2021). Others use hypernetworks (Raychaudhuri et al., 2022; Aich et al., 2023) which learn to generate MTL architectures conditioned on user preferences. While our method also models task relationships and capacity allocation, it does so without architecture search, relying solely on static modulation via normalization layers.

**Mixture-of-Experts (MoE)** methods address task interference by dynamically routing inputs to specialized experts, enabling flexible capacity allocation among tasks (Ma et al., 2018; Hazimeh et al., 2021; Tang et al., 2020). More recent work extends MoE designs to large-scale transformer architectures for vision and language tasks (Fan et al., 2022; Chen et al., 2023; Ye & Xu, 2023; Yang et al., 2024). Although effective, these methods rely on dynamic, per-sample routing that increases architectural and training complexity. In contrast, our approach provides a static and lightweight form of soft partitioning, achieving similar benefits with minimal changes to the wrapped backbone.

**Parameter-efficient fine-tuning (PEFT)** is a popular approach for adapting large pre-trained models without updating the full backbone. Single-task PEFT methods such as Adapters (He et al., 2021), BitFit (Zaken et al., 2022), VPT (Jia et al., 2022), Compacter (Karimi Mahabadi et al., 2021), and LoRA-style updates add small task-specific modules or low-rank layers while keeping most weights frozen. Extending

these ideas to MTL requires managing several task-specific adapters at once. Recent PEFT-MTL methods address this by generating adapter weights through hypernetworks or decompositions, as in HyperFormer (Mahabadi et al., 2021), Polyhistor (Liu et al., 2022b), and MTLoRA (Agiza et al., 2024). However, these methods still rely on additional task-specific capacity, which parallels traditional soft-parameter sharing and scales poorly with the number of tasks. In contrast, we modulate the shared capacity directly through BN, without adding new feature extractors.

**Domain-specific normalization** has become a common technique in settings with domain shift, where shared BatchNorm fails because domains have different input distributions. In these cases, separate BN statistics or layers are required to maintain stable normalization (Li et al., 2016; Zajac et al., 2019; Chang et al., 2019). The same motivation appears in several areas: In meta-learning, TaskNorm (Bronskill et al., 2020) adapt BN statistics per episode to handle changes in input distribution. In continual learning, CLBN (Xie et al., 2023) store task-specific BN parameters to avoid catastrophic forgetting from normalization drift. In conditional or multi-modal models, BN and LayerNorm is adjusted to match modality-specific statistics (Michalski et al., 2019; Zhao et al., 2024). In multi-domain MTL (Bilen & Vedaldi, 2017; Mudrakarta et al., 2019; Wallingford et al., 2022; Deng et al., 2023), task-specific BN is used as an adapter for tasks from different domains. In contrast, our work targets single-domain MTL, where all tasks share the same input and normalization does not fail. In this case, task-specific BN is not needed for statistical correction. Instead, we focus on its affine parameters as a basis for task-specific feature modulation, and extend this idea with a reparameterization and optimization scheme tailored to reduce interference and allocate capacity.

**Multi-task optimization (MTO)** methods address task interference and dominance by balancing training dynamics in hard-shared networks. Loss-based approaches reweight task losses using criteria such as uncertainty (Kendall et al., 2017), gradient norms (Chen et al., 2018), or loss change rates (Liu et al., 2019; 2024), while gradient-based methods reduce conflict by modifying gradients (Chen et al., 2020; Yu et al., 2020; Wang et al., 2021) or solving multi-objective formulations for Pareto-optimal updates (Sener & Koltun, 2018; Navon et al., 2022; Liu et al., 2021a). Though effective, these often incur high computational cost during training. Recent work shows that simple scalarization strategies can match their performance at lower cost (Kurin et al., 2022; Xin et al., 2022). Our method is orthogonal, focusing on architectural design: despite not performing explicit reweighting, it exhibits loss-scale robustness by implicitly balancing tasks through its normalization-based structure.

## 3 BatchNorm and $\sigma$BatchNorm

Batch normalization is a cornerstone for deep CNNs due to its versatility, efficiency, and wide-ranging benefits, including improved training stability for faster convergence (Santurkar et al., 2018; Bjorck et al., 2018), regularization effects (Luo et al., 2019), and the orthogonalization of representations (Daneshmand et al., 2021). BN operates in two key steps - normalization and affine transformation:

$$BN(x; \gamma, \beta) = \gamma \hat{x} + \beta, \qquad \hat{x} = \frac{x - \mu_B}{\sqrt{\sigma_B^2 + \epsilon}} \tag{1}$$

The normalization step standardizes input activations using the mini-batch mean $\mu_B$ and variance $\sigma_B^2$, while the affine transformation applies channel-specific learnable parameters, $\gamma$ and $\beta$, to re-scale and shift the normalized activations. During inference, BN relies on population statistics collected during training via running estimates. When the test distribution differs from the training set, these statistics can become mismatched and significantly degrade model performance (Summers & Dinneen, 2020). Because of this, many BN variants aim to improve the normalization step itself by adjusting $\mu$ and $\sigma$ to handle distribution changes, domain shift, meta-learning episodes, or multi-modal inputs. For a survey on normalization approaches we refer to Huang et al. (2023).

In single-domain MTL, all tasks share the same input distribution, so the normalization component of BN does not need adjustment. Instead, we focus on the affine transformation post-normalization. These parameters represent only a small fraction of the network, yet they have substantial expressive power, as shown by studies demonstrating high performance when training BN alone (Frankle et al., 2021). In this work,

we build on a variation of BN originally introduced to determine feature importance in structured pruning, Sigmoid Batch Normalization (Suteu & Guo, 2022) replaces the affine transformation with a bounded scaler:

$$\sigma BN(x; \gamma) = \sigma(\gamma)\hat{x}, \qquad \sigma(\gamma) = \frac{1}{1 + e^{-\gamma}} \tag{2}$$

Using a single bounded scaler per feature has little impact on performance, but enables targeted regularization and improves interpretability. These properties make $\sigma$BN especially attractive for multi-task learning, where understanding how tasks share limited capacity is critical. In this setting, $\sigma(\gamma)$ acts as a static soft gate that can down-weight or disable features. This implicit static gating contrasts with soft-sharing models, which explicitly partition capacity, and MoE methods, which route features dynamically through task-specific gates. Furthermore, this formulation can be extended to other normalization layers (Ba et al., 2016), as we show in experiments on transformers. Using $\sigma$BN as the only task-specific component, we create a highly effective, parameter-efficient architecture that comes with a built-in framework to analyze capacity allocation, task relationships and feature importance.

## 4  Task-Specific $\sigma$BatchNorm Networks

TS$\sigma$BN networks are constructed by replacing every shared BN layer with task-specific $\sigma$BN layers, as illustrated in Figure 1. This design allows tasks to normalize and modulate the outputs of shared layers, with $\gamma_t \in \mathbb{R}^C$ a per-channel learnable vector and $\sigma$ applied elementwise:

$$TS\sigma BN(x; \gamma_t) = \sigma(\gamma_t)\hat{x}, \qquad \hat{x} = \frac{x - \mu_{B,t}}{\sqrt{(\sigma_{B,t})^2 + \epsilon}} \tag{3}$$

enabling better disentanglement of representations and reduced task interference. Unlike prior methods introducing additional task-specific capacity, TS$\sigma$BN keeps all feature extractors shared, preserving the multi-task learning inductive bias toward generalizable representations. While domain-specific BN has been used reactively in domain adaptation (Chang et al., 2019) to handle distribution shifts, our work is the first to use it proactively as a standalone mechanism in single-input scenarios.

**Discriminative Learning Rates**. We increase the learning rate of $\sigma$BN parameters by a fixed multiple ($\alpha_{\sigma BN} = 10^2$) relative to other parameters, allowing them to allocate filters before these undergo significant updates. This accelerates specialization and ensures capacity allocation occurs early in training. A further advantage of $\sigma$BN is its robustness to high learning rates: the sigmoid dampens gradients, making training stable across scales, whereas vanilla BN is more sensitive and requires careful tuning. The approach parallels transfer learning, where deeper layers are updated more aggressively to drive adaptation (Howard & Ruder, 2018; Vlaar & Leimkuhler, 2022). We provide ablations on the effect of higher learning rates. The full training loop is given in Algorithm 1.

**Task interference**. Conflicting gradients between tasks is a central challenge in MTL, often measured by negative cosine similarity (Zhao et al., 2018; Yu et al., 2020; Shi et al., 2023). Figure 2 (left) shows the gradient similarity distribution for shared parameters: in hard parameter sharing, the distribution is nearly uniform, meaning roughly half of all updates conflict. MTAN (Liu et al., 2019) partially alleviates this issue by introducing task-specific convolutions. In contrast, TS$\sigma$BN yields a sharper zero-centered distribution, indicating mostly orthogonal gradients. This mirrors optimization-based methods that explicitly enforce orthogonality (Yu et al., 2020; Suteu & Guo, 2019), yet TS$\sigma$BN achieves it through a lightweight architectural change. Figure 2 (middle) further supports this: on CelebA, task representations form well-separated clusters, illustrating reduced interference. A full analysis for all tasks is provided in Appendix A.

**Parameter Efficiency.** Task-Specific $\sigma$BN is highly parameter efficient since it does not introduce additional feature extractors like related soft parameter sharing architectures. At the extreme end, such as Single Task Learning or Cross-Stitch networks, the entire backbone is duplicated for each new task. TS$\sigma$BN on the other hand duplicates only $\sigma$BN layers, whose parameters comprise a fraction of the total model size. Figure 2

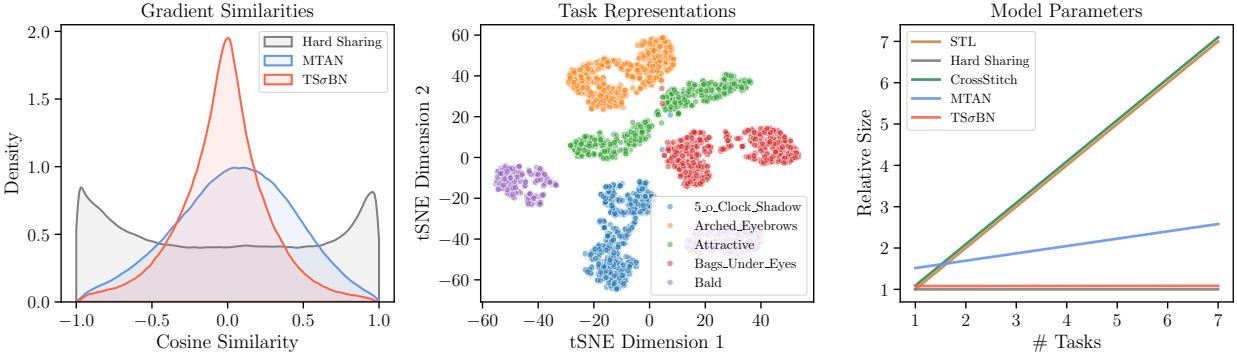

Figure 2: Left: Distribution of gradient cosine similarities between NYUv2 tasks in early stages of training. Middle: t-SNE visualization of encoder representations for the first five CelebA tasks. Right: Encoder parameter growth with numbers of tasks relative to a ResNet50 backbone. TS$\sigma$BN has a greater concentration of orthogonal gradients, produces separated task representations and negligible parameter growth.

(right) shows how different approaches scale with additional tasks. TS$\sigma$BN adds an insignificant amount of new parameters, allowing it to scale to any number of tasks.

## 5 MTL Analysis with TS$\sigma$BN

A key advantage of the TS$\sigma$BN design is the ability to quantify filter allocation through task-filter importance matrices. Since each $\sigma$BN layer introduces a dedicated scaling parameter $\gamma_{t,i}$ per task and filter, we construct a task-filter importance matrix $I \in \mathbb{R}^{T \times F}$, where each entry $I_{t,i}$ captures the importance task $t$ assigns to filter $i$. Applying the sigmoid function to the raw scaling parameters $I_{t,i} = \sigma(\gamma_{t,i})$ ensures that values remain within $[0, 1]$, facilitating interpretability and comparability across tasks, layers, and models. Using this representation, TS$\sigma$BN enables a principled analysis of MTL dynamics, including capacity allocation, task relationships, and filter specialization.

### 5.1 Capacity Allocation

One of the central challenges in multi-task learning is understanding how model capacity is allocated among competing tasks. The TS$\sigma$BN task-filter importance matrix $I$ can directly quantify the total capacity of a task $t$ as the normalized sum of the importances it assigns to filters $C_t = \frac{1}{F} \sum_{i=1}^{F} \sigma(\gamma_{t,i})$. This measure provides an overall assessment of the resources required for each task; however, it does not account for task relationships or shared capacity. A task with high absolute capacity does not necessarily imply it monopolizes filters, as it may rely heavily on shared generic filters.

We apply an orthogonal projection-based decomposition to differentiate between task-specific and shared capacity. Given the set of task importance vectors $\{I_1, I_2, ..., I_T\}$, we decompose each task's capacity into an independent component and a shared component. Let $A$ be the matrix formed by stacking all task importance vectors except $I_t$. The projection of $I_t$ onto the subspace spanned by the other tasks is given by the projection matrix $P_A$:

$$P_A I_t = A(A^T A)^{-1} A^T I_t, \tag{4}$$

The shared $\hat{I}_t = P_A I_t$ and independent $I_t^\perp = I_t - \hat{I}_t$ components of $I_t$ can therefore be defined so that $I_t^\perp$ is orthogonal to the subspace spanned by the other task importance vectors.

To derive a capacity decomposition consistent with the original measure, we define the independent and shared capacities as scaled versions of the total capacity:

$$C_t^{indep} = \frac{\|I_t^\perp\|_2}{\|I_t\|_2} C_t, \qquad C_t^{shared} = \frac{\|\hat{I}_t\|_2}{\|I_t\|_2} C_t. \tag{5}$$

Because in this formulation the components are orthogonal, the $L_2$ norm satisfies the Pythagorean theorem, yielding $C_t^2 = (C_t^{shared})^2 + (C_t^{indep})^2$. This guarantees that a task's total capacity is preserved while providing an interpretable split between shared and independent resource usage.

Using our framework, we analyze task capacity allocation after training as shown in Figure 3. For both SegNet and DeepLabV3 architectures, we find that most capacity is shared among tasks without a single task dominating. For a more detailed analysis on the effects of task difficulty and similarity on capacity allocation, we refer to Appendix E.

## 5.2    Task Relationships

A desirable feature for any multi-task learning model is the ability to derive task relationships, as this can help gauge interference between tasks and provide insights into the joint optimization process. To showcase this, we use the CelebA dataset, containing 40 binary facial attribute tasks, allowing us to explore complex task relationships and hierarchies via TS$\sigma$BN. Moreover, because these attributes are semantically interpretable (e.g., "Smiling", "Mouth Slightly Open"), they enable meaningful qualitative assessments of the learned relationships.

To derive task relationships we compute the pairwise cosine similarity between the task importance vectors $I_t \in \mathbb{R}^F$, yielding a $T \times T$ similarity matrix, with

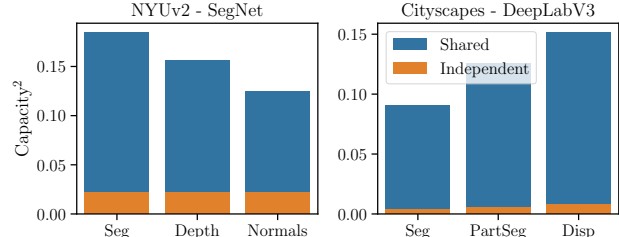

Figure 3: Decomposed task capacity into shared and independent components using the TS$\sigma$BN framework. In all standard scenarios, tasks share most capacity without signs of dominance.

values ranging from 0 (orthogonal filter usage) to 1 (indicating identical usage). We use this as the basis for constructing distance matrices to identify task clusters and hierarchical relationships that reflect the model's capacity allocation. To assess the stability of the task relationships derived from our model, we focus on the consistency of task hierarchies across multiple training runs. Specifically, we evaluate the similarity matrices obtained from seven independently trained models with different intializations. We compute the pairwise Spearman rank correlation between similarity matrices to determine whether the relative task orderings are robust to such variations. Our results show that the task hierarchies are highly stable, with an average Spearman correlation of 0.8 across all model pairs.

We further assess the resulting relationships by aggregating the representative task clusters from the seven runs, via co-occurrence matrices and hierarchical clustering. The identified clusters exhibit semantic coherence, suggesting a correlation with the spatial proximity of facial attributes. For instance, tasks related to hair characteristics (e.g., Bangs, Blond Hair) form a distinct cluster. In contrast, facial hair attributes (e.g. Goatee, Mustache) are grouped separately.

To test whether this structure is unique to TS$\sigma$BN, we construct an analogous task-filter importance matrix for a vanilla hard-parameter-sharing baseline by ablating each filter individually and measuring the resulting per-task performance drop on the full validation set. Unlike the TS$\sigma$BN matrix, which is read directly from trained parameters, this requires a full evaluation pass per filter. Using the same clustering pipeline we observe that the strongest pairs are recovered by both methods and appear as strict subsets of TS$\sigma$BN's clusters, indicating that the dominant structure reflects real shared filter usage. The broader cluster structure however is not stable and has only a mean Spearman rank correlation 0.27. At that level only the strongest pairs survive co-occurrence aggregation, while the larger semantic groups collapse to noise. More details about the procedure and resulting task clusters can be found in the Appendix C.

## 5.3    Filter Groups

A different way to analyze multi-task learning is from an individual filter perspective. Using the task-filter matrix, we can gauge each task's reliance on a filter to determine if the resource is specialized or generic. We define a filter as specialized for a particular task if its normalized task-filter importance exceeds a threshold $\tau$.

We set $\tau = 0.5$ to signify that the filter predominantly contributes to a single task rather than being shared among multiple tasks. Formally, let $\sigma(\gamma_{t,i})$ denote the importance of filter $i$ for task $t$. A filter $i$ is deemed specialized for task $t'$ if $\sigma(\gamma_{t',i})/\sum_t^T \sigma(\gamma_{t,i}) > \tau$.

We prune the top 200 most important filters per task to test our definitions of specialization and importance. If accurate, removing a task's specialized filters should degrade its performance more than others. Figure 4 (right) confirms this: diagonal elements, representing self-impact, show significantly larger drops than off-diagonals, supporting our hypothesis.

Next, we examine where specialized filters occur across the network. Figure 4 (left) shows the percentage of specialized filters per layer from different runs. Specialization increases with network depth, indicating that early layers are more shared while deeper layers become task-specific. This mirrors findings in single-task learning (Yosinski et al., 2015), where lower layers encode general features, and aligns with branching-based NAS heuristics (Bruggemann et al., 2020; Vandenhende et al., 2020a; Guo et al., 2020), which assign specialized layers to later stages. Our method for quantifying specialization and task similarity offers an alternative perspective for NAS strategies.

These analyses are descriptive rather than causal: the task-filter importance matrix reflects how the trained model has allocated capacity, not why a particular allocation emerged. Still, the framework

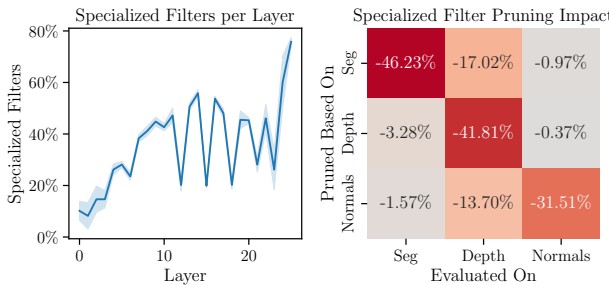

Figure 4: Left: Percentage of specialized filters per layer in a TS$\sigma$BN SegNet. Specialization increases with depth. Right: Performance drop across tasks (columns) after pruning filters based on their primary specialization (rows).

provides a useful diagnostic lens for MTL dynamics - a tool to look under the hood when something does not work, particularly when task relationships are not known a priori. Worth noting also that this analysis is performed on models trained from scratch with full task-specific normalization layers, and may behave differently on fine-tuned models where $\sigma$BN/$\sigma$LN parameters are introduced over already-converged features.

## 6 Experiments

We evaluate TS$\sigma$BN across a wide range of MTL settings - covering three CNN (from scratch and pretrained) and two vision transformer architectures over four standard MTL datasets: NYUv2 (Silberman et al., 2012), Cityscapes (Cordts et al., 2016), CelebA (Liu et al., 2015) and PascalContext (Chen et al., 2014). We follow established protocols from prior work (Liu et al., 2019; Ban & Ji, 2024; Lin & Zhang, 2023; Yang et al., 2024; Agiza et al., 2024) for training, evaluation, and metric reporting. TS$\sigma$BN achieves comparable or superior performance to related and state-of-the-art methods while maintaining better resource efficiency. We refer to Appendix F for additional details on TS$\sigma$BN integration, datasets, protocols and baselines.

**CNNs.** We evaluate TS$\sigma$BN in two settings: models trained from scratch and initialized from pretrained backbones. For models trained from scratch, we follow standard protocols on NYUv2 (3-task) using SegNet (Badrinarayanan et al., 2017) as in Liu et al. (2019), and on Cityscapes (3-task) using DeepLabV3 (Chen, 2017) following Liu et al. (2022a). For CelebA, which contains 40 binary classification tasks, we adopt the CNN architecture used in Liu et al. (2024); Ban & Ji (2024). For pretrained CNNs, we use LibMTL (Lin & Zhang, 2023) and DeepLabV3 with a pretrained ResNet50 backbone on NYUv2 (3-task) and Cityscapes (2-task). This allows comparison to a wide range of recent MTL baselines under a consistent framework.

**Vision Transformers.** We evaluate TS$\sigma$BN on two transformer-based MTL setups that reflect current state-of-the-art: MoE-style modulation, and parameter-efficient adapter-based methods. Both settings use pretrained Vision Transformer backbones with CNN based fusion or downsampling modules before task-specific decoders. For recent MoE MTL methods we follow the MLoRE protocol (Yang et al., 2024) on PascalContext (5-task). We use a pretrained ViT-S backbone (Dosovitskiy et al., 2021) and fine-tune the entire model. We

Table 1: Comparison of encoder-based soft-sharing architectures on NYUv2 (3-task SegNet), Cityscapes (3-task DeepLabV3), and CelebA (40-task CNN) trained from scratch. All results are mean$_{(std)}$ over three random seeds TS$\sigma$BN significantly outperforms on NYUv2 and CelebA, is competitive on Cityscapes, while maintaining the lowest parameter count.

| Method | NYUv2 | | | | | Cityscapes | | | | | CelebA | | |
|---|---|---|---|---|---|---|---|---|---|---|---|---|---|
| | #P | Seg↑ | Depth↓ | Norm↓ | Δ% | #P | Seg↑ | P.Seg↑ | Disp↓ | Δ% | #P | F1↑ | Δ% |
| STL | 1.00 | 41.45 | 0.580 | 23.80 | 0.00 | 1.00 | 56.61 | 53.95 | 0.841 | 0.00 | 1.00 | 68.21 | 0.00 |
| HPS | 0.33 | 42.17 | 0.502 | 26.63 | +1.07$_{(0.66)}$ | 0.60 | 55.03 | 51.92 | 0.796 | -0.39$_{(0.75)}$ | 0.03 | 67.06 | -1.69$_{(0.70)}$ |
| CS | 1.00 | 41.77 | 0.492 | 26.15 | +1.98$_{(1.55)}$ | 1.00 | 56.73 | 53.89 | 0.781 | +2.43$_{(0.08)}$ | 1.01 | 65.57 | -3.86$_{(1.74)}$ |
| MTAN | 0.59 | 43.12 | 0.508 | 25.44 | +3.14$_{(0.41)}$ | 0.78 | 55.83 | 52.61 | 0.799 | +0.39$_{(0.65)}$ | 0.39 | 59.49 | -12.78$_{(2.84)}$ |
| TSBN | 0.33 | 43.47 | 0.494 | 25.32 | +4.42$_{(0.73)}$ | 0.61 | 56.10 | 52.82 | 0.806 | +0.40$_{(0.11)}$ | 0.03 | 67.17 | -1.52$_{(1.13)}$ |
| **TS$\sigma$BN** | **0.33** | 43.75 | 0.484 | 24.09 | **+6.93**$_{(0.23)}$ | **0.60** | 56.45 | 53.26 | 0.814 | +0.57$_{(0.19)}$ | **0.03** | 69.45 | **+1.81**$_{(0.73)}$ |

Table 2: Comparison of various multi-task architectures within the LibMTL framework using DeepLabV3 with a pre-trained ResNet-50 backbone on NYUv2 (3-task) and CityScapes (2-task). TS$\sigma$BN achieves the best overall performance while being the most parameter-efficient. Cityscapes stds are computed from re-runs of all methods; NYUv2 baseline values are reported as published in Lin & Zhang (2023).

| Method | NYUv2 | | | | | | CityScapes | | | | |
|---|---|---|---|---|---|---|---|---|---|---|---|
| | #P | #F | Seg↑ | Depth↓ | Normal↓ | Δ% | #P | #F | Seg↑ | Depth↓ | Δ% |
| HPS | 1.00 | 1.00 | 53.93 | 0.3825 | 23.57 | 0.00 | 1.00 | 1.00 | 69.81 | 0.0125 | 0.00 |
| CS | 1.65 | 1.69 | 53.44 | 0.3818 | 23.15 | +0.35 | 1.42 | 1.44 | 69.97 | 0.0123 | +0.55 $_{(0.76)}$ |
| MMOE | 1.35 | 1.34 | 53.14 | 0.3876 | 23.02 | -0.15 | 1.42 | 1.44 | 69.81 | 0.0126 | -0.43 $_{(1.33)}$ |
| MTAN | 1.28 | 1.56 | 54.64 | 0.3771 | 23.12 | +1.55 | 1.29 | 1.48 | 70.62 | 0.0125 | +0.49 $_{(0.79)}$ |
| CGC | 2.01 | 2.03 | 53.27 | 0.3914 | 22.14 | +0.84 | 1.85 | 1.88 | 69.75 | 0.0125 | -0.12 $_{(1.60)}$ |
| PLE | 2.41 | 2.71 | 52.75 | 0.3943 | 22.10 | +0.32 | 1.95 | 2.32 | 69.30 | 0.0129 | -2.02 $_{(0.26)}$ |
| LTB | 1.65 | 1.69 | 52.58 | 0.3828 | 23.31 | -0.49 | 1.42 | 1.44 | 69.81 | 0.0125 | -0.35 $_{(0.77)}$ |
| DSelect-k | 1.38 | 1.34 | 53.75 | 0.3802 | 23.18 | +0.64 | 1.44 | 1.44 | 69.67 | 0.0124 | +0.26 $_{(0.27)}$ |
| TSBN | 1.00 | 1.69 | 53.44 | 0.3761 | 23.01 | +1.04 | 1.00 | 1.44 | 69.89 | 0.0124 | +0.38 $_{(0.38)}$ |
| **TS$\sigma$BN** | **1.00** | 1.69 | 53.78 | 0.3735 | 22.31 | **+2.48** | **1.00** | 1.44 | 70.17 | 0.0123 | **+0.85** $_{(0.45)}$ |

also evaluate TS$\sigma$BN on the MTLoRA benchmark (Agiza et al., 2024), which focuses on parameter-efficient MTL. This setup uses a partially frozen Swin-T (Liu et al., 2021c) backbone on PascalContext (4-task). We compare against a wide range of LoRA and adapter based models reported in MTLoRA. To showcase compatibility we also evaluate TS$\sigma$BN with added task-generic (shared) LoRA($r = 16$) adapters.

**Multi-task evaluation**. Following Maninis et al. (2019) to evaluate a multi-task model, we compute the average per-task performance gain or drop relative to a baseline $B$ specified in the top row of the results tables. $\Delta m\% = \frac{1}{T} \sum_{t=1}^{T} (-1)^{\delta_t} \frac{M_{m,t} - M_{B,t}}{M_{B,t}} \times 100$, where $M_{m,t}$ is the performance of a model $m$ on a task $t$, and $\delta_t$ is an indicator variable for metrics that should be minimized. All results are presented as an average over three independent runs, and where available we also add the standard deviation in parenthesis. Additionally, we report parameters (P) and FLOPs (F) relative to the baseline.

**Baselines.** Across all experiments we compare TS$\sigma$BN to a set of standard and protocol-specific multi-task baselines. The most common reference points are Single-Task Learning (STL), which trains a separate model for each task, and Hard Parameter Sharing (HPS), which shares the entire backbone with equal task weights. We also include TSBN, the multi-task equivalent of domain-specific BN, which simply duplicates BN layers without our reparameterization and optimization. Each experimental setting includes additional baselines that follow the protocol and architecture family, reflecting standard practice in prior work and ensuring fair comparisons. For completeness, we report results for optimization-based methods in Appendix G.

## 6.1 Results

Across all experiments, TS$\sigma$BN delivers consistent performance gains while being more parameter efficient. On randomly initialized CNNs in Table 1, TS$\sigma$BN achieves the best results on NYUv2 (+6.93%) and CelebA (+1.81%), with competitive performance on Cityscapes, all at the lowest parameter cost. Notably, soft parameter sharing methods underperform the STL baseline on CelebA, highlighting their poor scalability to many tasks, whereas TS$\sigma$BN remains robust. On pretrained CNNs within LibMTL in Table 2, TS$\sigma$BN achieves the strongest overall performance on both NYUv2 (+2.48%) and Cityscapes (+0.85%), outperforming all MTL baselines, including MoE approaches, while remaining lightweight.

Table 3: PascalContext results for MoE-style models using a pretrained ViT-S backbone. TS$\sigma$BN delivers best results using fewer parameters.

| Method | Seg. mIoU↑ | Parts. mIoU↑ | Sal. maxF↑ | Norm. mErr↓ | Bdry. odsF↑ | #F (G) | #P (M) |
|---|---|---|---|---|---|---|---|
| M³ViT | 72.80 | 62.10 | 66.30 | 14.50 | 71.70 | 420 | 42 |
| Mod-Squad | 74.10 | 62.70 | 66.90 | 13.70 | 72.00 | 420 | 52 |
| TaskExpert | 75.04 | 62.68 | 84.68 | 14.22 | 68.80 | 204 | 55 |
| MLoRE | 75.64 | 62.65 | 84.70 | 14.43 | 69.81 | 72 | 44 |
| TSBN | 75.95 | 63.33 | 84.65 | 14.16 | 68.05 | 214 | 29 |
| **TS$\sigma$BN** | **77.12** | **64.73** | **85.24** | 14.04 | 70.00 | 214 | **29** |

On pre-trained transformers with ViT-S in Table 3, TS$\sigma$BN surpasses state-of-the-art methods, such as M³ViT, Mod-Squad, and MLoRE, while using fewer parameters. Relative to other parameter-efficient fine-tuning approaches in Table 4 TS$\sigma$BN offers the best performance relative to its trainable parameter count. Adding shared capacity via LoRA($r = 16$) adapters further improves performance.

To better characterize the parameter-efficiency frontier, and to disentangle increased capacity from that of individual task specialization, we sweep two LoRA variants across ranks $r \in \{4, 8, 16, 32, 64\}$: *Shared LoRA*, where a single adapter is shared across tasks, and *Per-Task LoRA*, where each task has its own. The full results are shown in Appendix H. Shared LoRA never outperforms STL regardless of capacity, while Per-Task LoRA crosses zero at $r = 32$ and only matches TS$\sigma$BN at $r = 64$, with $\sim 2\times$ the trainable parameters. The gap confirms that modulating shared capacity is more parameter-efficient than adding task-specific.

We note that even the simpler TSBN baseline delivers competitive performance out of the box, suggesting that complex architectures may be unnecessarily over-engineered. Overall, TS$\sigma$BN achieves the best balance of accuracy, efficiency, and simplicity, consistently outperforming specialized MTL architectures across CNNs and transformers, while scaling to many-task regimes.

## 7 Ablations

### 7.1 Discriminative Learning Rates

We analyze the impact of different learning rate multipliers applied to the $\sigma$BN layers, focusing on their effect on the distribution of scaling parameters $\gamma_t$ and overall model performance. Figure 5 illustrates how varying the $\alpha_{BN}$ multiplier influences the distribution of $\sigma(\gamma_t)$ values across all filters. A more detailed task-wise breakdown is provided in the Appendix. Higher learning rates induce more significant parameter variance, increasing their expressivity. Since $\sigma(\gamma_t)$ is initialized at 0.5, lower learning rates result in minimal divergence,

Table 4: PEFT baselines using a Swin-T backbone on Pascal-Context sorted by number of trainable parameters. TS$\sigma$BN and its combination with LoRA($r = 16$) deliver the best performance relative to their size.

| Method | Seg. mIoU↑ | Parts mIoU↑ | Sal. mIoU↑ | Norm. mErr↓ | Δm (%) | #P (M) |
|---|---|---|---|---|---|---|
| STL | 67.21 | 61.93 | 62.35 | 17.97 | 0 | 112.62 |
| HyperFormer | 71.43 | 60.73 | 65.54 | 17.77 | 2.64 | 72.77 |
| MTL-Full FT | 67.56 | 60.24 | 65.21 | 16.64 | 2.23 | 30.06 |
| Adapter | 69.21 | 57.38 | 61.28 | 18.83 | -2.71 | 11.24 |
| Polyhistor | 70.87 | 59.15 | 65.54 | 17.77 | 2.34 | 8.96 |
| MTLoRA(r=16) | 68.19 | 58.99 | 64.48 | 17.03 | 1.35 | 4.95 |
| VL-Adapter | 70.21 | 59.15 | 62.29 | 19.26 | -1.83 | 4.74 |
| **TS$\sigma$BN(r=16)** | **70.00** | **58.01** | **63.89** | **16.85** | **1.63** | **4.25** |
| VPT-deep | 64.35 | 52.54 | 58.15 | 21.07 | -10.85 | 3.43 |
| MTLoRA+(r=8) | 68.54 | 58.30 | 63.57 | 17.41 | 0.29 | 3.15 |
| **TS$\sigma$BN** | **69.38** | **57.46** | **63.74** | **17.00** | **0.91** | **3.08** |
| TSBN | 69.12 | 57.00 | 62.76 | 17.56 | -0.54 | 3.08 |
| LoRA | 70.12 | 57.73 | 61.90 | 18.96 | -2.17 | 2.87 |
| BitFit | 68.57 | 55.99 | 60.64 | 19.42 | -4.60 | 2.85 |
| Compacter | 68.08 | 56.41 | 60.08 | 19.22 | -4.55 | 2.78 |
| Compacter++ | 67.26 | 55.69 | 59.47 | 19.54 | -5.84 | 2.66 |
| VPT-shallow | 62.96 | 52.27 | 58.31 | 20.90 | -11.18 | 2.57 |

with $\alpha_{\sigma BN} = 1$ being excluded as it shows almost no differentiation between tasks. At $\alpha_{\sigma BN} = 100$, we see a substantial spread in $\sigma(\gamma_t)$ values across the full $[0, 1]$ range, allowing tasks to choose and specialize on subsets of filters. However, an extreme learning rate of $\alpha_{\sigma BN} = 10^3$ leads to a highly polarized distribution, where filter importances collapse to a binary mask, effectively enforcing a hard-partitioning regime. These findings highlight how BN learning rates control the degree of task-specific capacity allocation, influencing both representation disentanglement and network adaptability.

We further analyze the impact of different learning rate multipliers on the MTL performance in Table 6. For TSBN, moderate multipliers yield small gains, but performance collapses at high rates. In contrast, $\sigma$BN consistently benefits from larger multipliers across values, indicating that sigmoid activation is essential both for unlocking greater improvements and for robustness.

## 7.2 Robustness to Loss Scales

A well-known challenge in multi-task learning is the discrepancy in loss scales and, consequently, gradient magnitudes across tasks, which can lead to task dominance and suboptimal performance. Many existing approaches rely on manual tuning or specialized optimization strategies for dynamic weighting. Our method is highly robust to perturbations of loss scales without any additional changes.

To evaluate the robustness of our method to loss weight perturbations, we conduct a series of experiments on NYUv2 by varying the weight of each task. Specifically, we scale each task loss by factors of $\{0.5, 1.5, 2.0\}$ while maintaining the default weight of 1.0 for the remaining tasks. The distribution of relative performances under these perturbations is visualized in Figure 5. TS$\sigma$BN shows the lowest variance under loss scale perturbations, indicating robustness to task dominance and improved optimization stability.

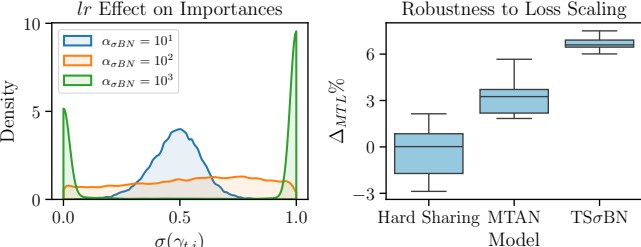

Figure 5: Effect of BN-specific learning rate multipliers on the $\sigma(\gamma_t)$ filter importance distribution (left) and relative performance under loss scale perturbations (right).

| $\alpha_{\sigma BN}$ | $10^0$ | $10^1$ | $10^2$ | $10^3$ |
|---|---|---|---|---|
| TSBN | +4.09% | +4.80% | +4.42% | -2.96% |
| TS$\sigma$BN | +4.02% | +5.67% | +6.93% | +4.33% |

Figure 6: Impact of different BN specific learning rate multipliers on the performance of TSBN and TS$\sigma$BN relative to STL on NYUv2.

## 8 Conclusion

We present TS$\sigma$BN, a simple soft-sharing mechanism for multi-task learning that relies only on task-specific normalization layers. Using a sigmoid-gated reparameterization and discriminative learning rates, our method turns normalization layers into stable and expressive tools for capacity allocation and interference reduction. Across convolutional and transformer architectures, TS$\sigma$BN achieves competitive or superior performance while using substantially fewer parameters. Notably, on the Transformer benchmarks evaluated, it matches or outperforms state-of-the-art MoE-style and PEFT-based MTL methods without adding routing modules, experts, or adapters. The learned gates also provide a direct view of model behavior, yielding interpretable measures of capacity allocation, filter specialization, and task relationships.

Looking broadly at the MTL landscape our architecture focused method is complementary to optimization based approaches. The two families trade off along different axes: MTO methods scale naturally with task count but rely on external solvers that become prohibitive for larger models, whereas parameter-efficient architectural methods like TS$\sigma$BN scale well with model size but are bounded by what their modulation mechanism can express.

*Limitations and future directions.* Although our work shows that aggressive simplification can be surprisingly effective for MTL, this minimalism introduces boundaries and tradeoffs that suggest natural extensions. The bounded gate $\sigma(\gamma_t) \in (0, 1)$ keeps tasks on a comparable filter-importance scale and stabilizes training, but

by construction it prevents any task from amplifying a filter beyond its native magnitude; settings where tasks have significantly different optimal feature scales may benefit from a less constrained parameterization. Our analysis framework also reveals when tasks share filters but does not actively shape that allocation. For highly related tasks, this could lead to redundant filter usage, making an explicit dissimilarity regularizer a natural way to encourage partitioning and diversity. Finally, our scope is deliberately single-domain MTL; multi-domain settings are already addressed by conditional BN methods (Xie et al., 2023; Chang et al., 2019; Deng et al., 2023), and we expect the mechanism to remain effective there.

Overall, our results show that lightweight, normalization-driven designs can replace much heavier mechanisms while offering clearer interpretability. We hope this encourages a reevaluation of complexity in MTL and promotes simple, transparent alternatives.

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

## A    Interference

To further investigate task interference, we expand on the analysis presented in Section 4 and provide a more comprehensive view of gradient conflicts across all task pairs for NYUv2. Specifically, in figure 7 we plot the distribution of cosine similarities between gradients for every task pair across the shared parameters of the SegNet backbone.

In addition to the methods discussed in the main paper, we include Task-Specific Batch Normalization (TSBN) as a baseline. Interestingly, TSBN alone is sufficient to induce a mode around orthogonality, demonstrating that normalization can already reduce some degree of task interference. However, incorporating $\sigma$BN significantly amplifies this effect, further increasing the number of near-orthogonal gradients and reducing interference. This highlights the role of $\sigma$BN in not only mitigating conflicts but also improving gradient disentanglement across tasks.

It is important to note that the presented gradient distributions are measured after one epoch of training over the training set. As training progresses, we observe that the differences between methods become less pronounced. Regardless of the initial distribution, all approaches gradually converge toward a bell-shaped distribution centered around orthogonality. This suggests that while early-stage interference may impact optimization dynamics, multi-task models eventually adjust to reduce conflicts over time.

A notable exception is observed in MTAN, which produces more aligned gradients specifically for the semantic segmentation and surface normal estimation task pair. Despite this alignment, we do not observe a corresponding performance gain. This suggests that while reducing conflicts is beneficial, not all aligned gradients lead to improved task synergy, underscoring the notion that mitigating interference alone does not guarantee optimal performance.

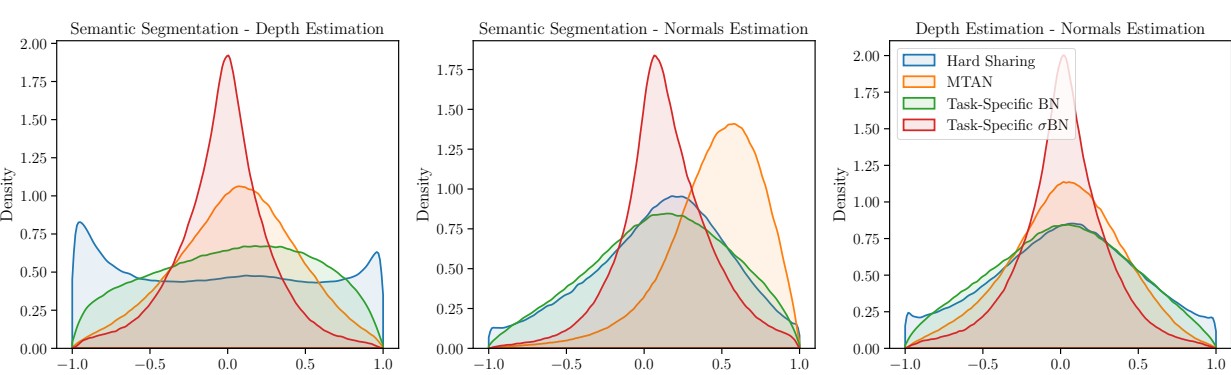

Figure 7: Distribution of gradient cosine similarities between all task pairs on the NYUv2 dataset using a SegNet backbone.

## B    Disentangled Task Representations

We extend Figure 2 from Section 4 by visualizing encoder representations for all 40 tasks in the CelebA setting. As before, we use t-SNE to project the high-dimensional representations into a more interpretable space. Each data point is assigned representations for every task due to the nature of the soft parameter sharing paradigm, resulting in multiple embeddings per sample. In Figure 8, we observe that most tasks form well-separated clusters, though a few outliers exhibit some degree of overlap.

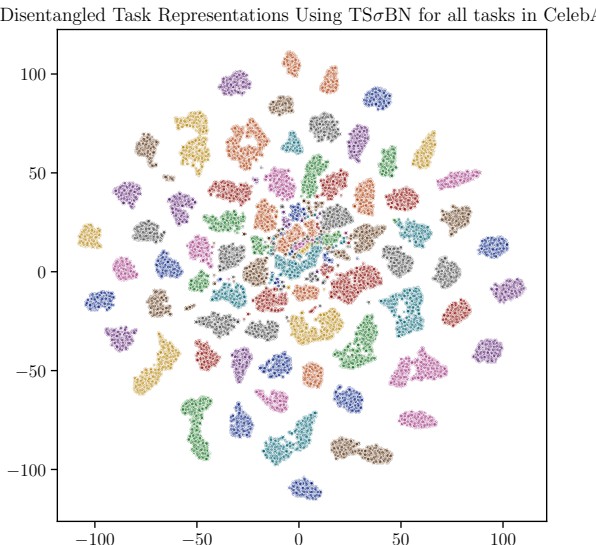

Figure 8: t-SNE visualization for all task representations for 1000 inputs from the CelebA dataset.

## C  Robust Task Relationships

**Setup.** We utilize the CelebA dataset to identify relationships and hierarchies among the 40 binary classification tasks of facial attributes. Given a per-seed task importance matrix $I \in \mathbb{R}^{T \times F}$, we compute pairwise cosine similarities between task vectors, producing a task similarity matrix $S = \left[ \frac{I_i \cdot I_j}{\|I_i\| \, \|I_j\|} \right]_{i,j \in \mathcal{T}}$. We train seven independent runs with different initializations and aggregate the resulting clusterings into a co-occurrence matrix that captures the frequency with which each pair of tasks appears in the same cluster. Hierarchical clustering on this matrix yields the representative clusters reported below. Cross-run stability is measured by the average Spearman rank correlation between similarity matrices across the seven runs. The same pipeline is applied to both TS$\sigma$BN and the baseline described later, with the only difference being how $I$ is obtained.

**TS$\sigma$BN.** The importance matrix is read directly from the trained $\sigma$BN parameters as $I_{t,i} = \sigma(\gamma_{t,i})$, at no additional cost beyond training. We find the resulting relationships to be highly stable, with an average Spearman rank correlation of **0.80** across the seven runs. The representative clusters exhibit apparent semantic coherence, as shown in Table 5. Since these clusters are derived from filter-usage based relationships, tasks grouped tend to rely on similar specialized filters within the network. This suggests that the model internally organizes tasks based on shared feature representations. Notably, the clustering patterns appear to correlate with the spatial proximity of facial attributes. For instance, tasks related to hair characteristics (e.g., Bangs, Blond Hair) form a distinct cluster. In contrast, facial hair attributes (e.g., Goatee, Mustache) are grouped separately, indicating that the network leverages localized feature detectors. This spatial coherence reinforces the idea that task relationships emerge from shared activations of filters sensitive to specific facial regions, reflecting the model's ability to capture both semantic and structural commonalities across tasks.

**Hard-sharing baseline.** To test whether this structure is unique to TS$\sigma$BN, we construct an analogous importance matrix for a vanilla hard-parameter-sharing model on the same task. For each filter $i$, we ablate it individually and measure the resulting per-task F1 drop on the full validation set, with $I_{t,i} = \max(F1_t^{\text{base}} - F1_t^{\text{pruned},i}, 0)$. This procedure produces an importance matrix of the same shape as TS$\sigma$BN's but requires a full evaluation pass per filter and scales as $\mathcal{O}(F \cdot |\text{val}|)$, in contrast to the zero-cost readout from $\sigma$BN parameters.

The baseline recovers the strongest task pairs identified by TS$\sigma$BN, shown in bold in Table 5: (Bags Under Eyes, Big Nose), (Heavy Makeup, Wearing Lipstick), (High Cheekbones, Smiling), and (Chubby, Double Chin). Each pair appears as a strict subset of a TS$\sigma$BN cluster, indicating that the dominant structure reflects real shared filter usage rather than an artifact of $\sigma$BN. The broader cluster structure and its stability do not

| # | Attributes |
|---|---|
| 1 | **High Cheekbones**, Mouth Slightly Open, **Smiling** |
| 2 | Bangs, Black Hair, Blond Hair, Brown Hair, Gray Hair, Straight Hair, Wavy Hair, Wearing Hat |
| 3 | Attractive, **Bags Under Eyes**, **Big Nose**, Young |
| 4 | Bald, **Chubby**, **Double Chin**, Receding Hairline, Wearing Necktie |
| 5 | Blurry, **Heavy Makeup**, Male, Pale Skin, **Wearing Lipstick** |
| 6 | 5 o'Clock Shadow, Goatee, Mustache, No Beard, Sideburns |
| 7 | Arched Eyebrows, Bushy Eyebrows, Narrow Eyes |
| 8 | Eyeglasses, Rosy Cheeks |
| 9 | Big Lips, Oval Face, Pointy Nose |
| 10 | Wearing Earrings, Wearing Necklace |

Table 5: Clusters of attributes extracted from a TS$\sigma$BN model trained on the 40-task CelebA dataset. Task relationships correlate with the spatial proximity of facial features, suggesting that the model organizes tasks based on localized filter activations. Bolded attributes mark the strongest task pairs also recovered by the hard-parameter-sharing baseline via single-filter ablation; each baseline pair sits inside a larger TS$\sigma$BN cluster.

transfer: the baseline reaches a mean Spearman rank correlation of only **0.27** across seeds, approximately $3\times$ less stable than TS$\sigma$BN. At that level only the strongest pairs survive co-occurrence aggregation, while the larger semantic groups (hair color, beard, feminine-face, adornment) collapse to noise. The pairs the baseline recovers are the stable core of the structure TS$\sigma$BN resolves around them.

# D  Discriminative Learning Rates

We extend the ablation study from Section 7.1, investigating the impact of discriminative learning rates for $\sigma$BN layers. Specifically, we apply a higher learning rate to BN parameters, allowing them to adapt more rapidly to the shared convolutional layers before those layers undergo significant updates. This adjustment is controlled by a multiplier applied to the model's base learning rate.

In this more detailed analysis, we examine the importance distributions of filters per task across different learning rate multipliers. Figure 9 presents the resulting distributions for four multiplier values: $10^0, 10^1, 10^2, 10^3$. As the multiplier increases, the variance of filter importance distributions grows, leading to progressively softer filter allocations. At a multiplier of 1, BN parameters remain close to their initialization, resulting in near-uniform filter sharing across tasks, similar to hard parameter sharing. On the opposite extreme, a multiplier of $10^3$ effectively induces a binary filter mask, resembling a hard partitioning approach. Notably, $\sigma$BN plays a crucial role in stabilizing this process, as its sigmoid activation mitigates potential gradient explosion. We use $\alpha_{\sigma BN} = 10^2$ in all our experiments.

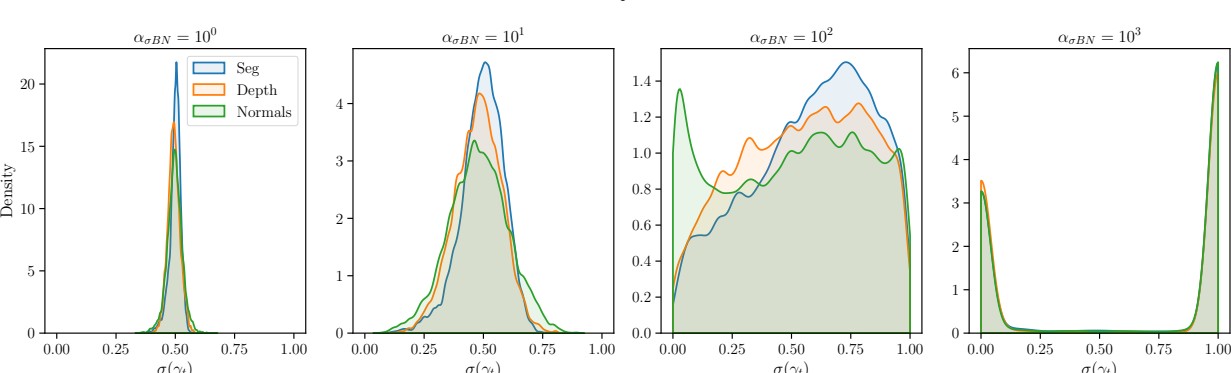

Figure 9: Detailed visualization of the effect of different learning rates on the distribution of task-specific $\sigma BN$ scaling parameters.

## E    Effects of Task Difficulty on Capacity Allocation

To further investigate MTL capacity allocation using the TS$\sigma$BN framework, we conduct a synthetic experiment designed to control task difficulty and relationships systematically. Specifically, we modify the NYUv2 dataset by removing the surface normals estimation task and replacing it with a noisy variant of the depth estimation task. We generate a family of datasets where the additional depth task is corrupted by Gaussian noise of increasing variance. Formally, given the original depth labels $D$, we construct synthetic tasks:

$$\tilde{D}_\xi = D + \mathcal{N}(0, \xi * \sigma_D^2), \tag{6}$$

where $\xi$ controls the level of corruption as a scaler of the original depth task's variance. Using TS$\sigma$BN, we analyze how model capacity is allocated between shared and task-specific components, as well as how task relationships change, by computing cosine similarity over task importance vectors.

In figure 10 we plot the decomposed task capacities and pairwise similarities for datasets with $\xi$ ranging between $[0, 3]$. As expected, when $\xi$ is low, the original and noisy depth tasks exhibit strong alignment, reinforcing high shared capacity. However, as $\xi$ increases, the similarity between the tasks decreases, and their filter allocations become more distinct, with independent capacity increasing. This aligns with our hypothesis that related tasks co-adapt to share resources, whereas unrelated tasks require greater specialization. Overall, this experiment highlights how TS$\sigma$BN automatically balances shared and independent capacity in response to increasing task difficulty and lower task similarity.

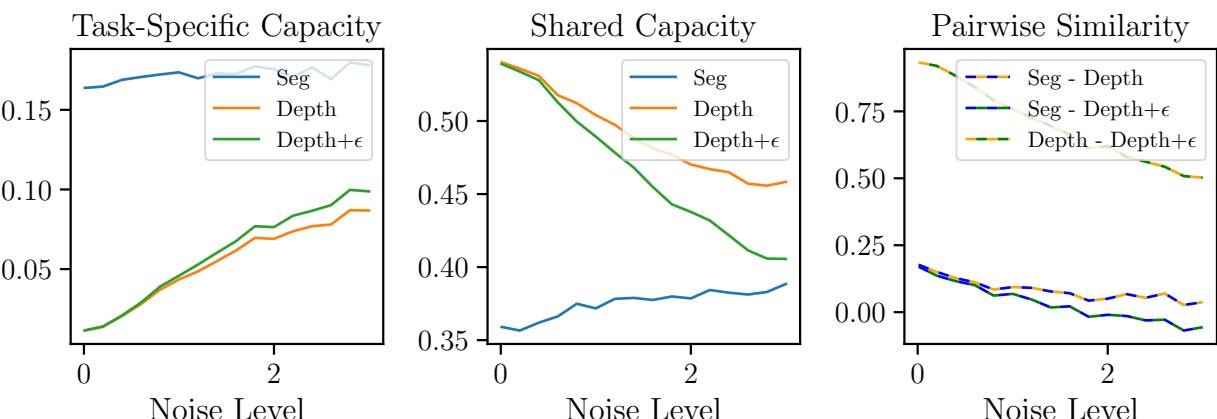

Figure 10: The effect of increasing task difficulty and decreasing similarity on capacity allocation in TS$\sigma$BN. For the noise scaling factor, a synthetic depth estimation task is generated with additive Gaussian noise. (Left) Independent task-specific capacities. (Middle) Shared capacity between tasks. (Right) Pairwise cosine similarity between task importance vectors.

## F   Experimental Settings

### F.1   Training Procedure

We use joint multi-task training: each batch contains samples from all tasks, per-task losses are summed, and gradients are accumulated through the shared backbone before a single optimizer step. The only deviation from standard MTL training is the discriminative learning rate applied to TS$\sigma$BN (and TS$\sigma$LN) parameters, which we make explicit below by separating the backbone update from the gate update.

---

**Algorithm 1** TS$\sigma$BN training loop

---

**Require:** shared backbone $f_\theta$, task heads $\{h_t\}_{t\in\mathcal{T}}$, TS$\sigma$BN params $\{\gamma_t\}_{t\in\mathcal{T}}$, base lr $\eta$, multiplier $\alpha_{\sigma BN}$
1:  **for** each training step **do**
2:      Sample batch $\{(x_i, \{y_{i,t}\}_{t\in\mathcal{T}})\}$
3:      $\mathcal{L} \leftarrow 0$
4:      **for** each task $t \in \mathcal{T}$ **do**
5:          $z_t \leftarrow f_\theta(x; \gamma_t)$ {forward through backbone with task $t$ params}
6:          $\hat{y}_t \leftarrow h_t(z_t)$
7:          $\mathcal{L} \leftarrow \mathcal{L} + \mathcal{L}_t(\hat{y}_t, y_t)$
8:      **end for**
9:      $\theta \leftarrow \theta - \eta \nabla_\theta \mathcal{L}$ {shared backbone and heads}
10:     $\gamma_t \leftarrow \gamma_t - \alpha_{\sigma BN} \eta \nabla_{\gamma_t} \mathcal{L} \quad \forall t \in \mathcal{T}$ {task-specific params with multiplier}
11: **end for**

---

**Hardware.** Experiments on NYUv2 and Cityscapes were run on an NVIDIA RTX 3090 GPU. Due to higher memory requirements, CelebA (40 tasks) and transformer-based models were trained on an NVIDIA A100 GPU.

### F.2   CNNs with Random Initialization

**NYUv2.** We follow the setup of Liu et al. (2019; 2024) for base architecture, training configuration, and evaluation metrics. A multi-task SegNet is used, with both encoder and decoder shared across tasks and lightweight task-specific heads composed of two convolutional layers. All methods are trained with Adam

($lr = 10^{-4}$), using a step schedule that halves the learning rate at epoch 100. Training runs for 200 epochs with a batch size of 4.

**Cityscapes.** Following Liu et al. (2022a), we use DeepLabV3 with a ResNet-50 backbone and task-specific ASPP decoders, which account for most of the parameters. Optimization is performed with SGD ($lr = 10^{-2}$, weight decay $= 10^{-4}$, momentum $= 0.9$) for 200 epochs using a CosineAnnealing scheduler and batch size of 4. For TS$\sigma$BN layers, weight decay is disabled.

**CelebA.** We adopt the configuration from Liu et al. (2024); Ban & Ji (2024), using a shared CNN backbone with task-specific linear classifiers. Models are trained for 15 epochs with Adam ($lr = 3 \times 10^{-4}$) and batch size 256.

### F.3 CNNs with Pretrained Weights

**Implementation.** Converting pretrained BN layers into $\sigma$BN depends on their weights. A network trained from scratch may learn a purely linear transformation, but converting an affine layer to linear is not possible unless $\beta = 0$. To avoid conversion shock, we copy the pretrained biases but keep them frozen during training. In ResNet-50 pretrained on ImageNet, most BN scale parameters ($\gamma$) fall within (0,1), allowing them to be represented by the sigmoid function. We therefore apply the inverse sigmoid to initialize $\sigma$BN scales, ensuring consistency with the pretrained distribution.

**NYUv2.** We follow the default LibMTL configuration (Lin & Zhang, 2023), reporting results of related methods as published. Models are trained with Adam ($lr = 10^{-4}$) for 200 epochs, using StepLR with $\gamma = 0.5$ at epoch 100 and a batch size of 4.

**Cityscapes.** Same as above, except the batch size is set to 16 due to memory constraints. All results, including related methods, are averaged over three random seeds for fair comparison.

### F.4 Vision Transformers

Following prior work, we extract intermediate representations from the transformer backbone and process them through a lightweight multi-scale fusion module. The module consists of four Conv–TS$\sigma$BN–GELU blocks shared across tasks, implemented as two 1×1 convolutions for channel adjustment squeezing two 3×3 convolutions with width 512; decoder inputs have width 196. In the ViT patch embedding, we replace the normalization with a $\sigma$LN layer. All remaining LNs in the backbone are converted to TSLN, since their pretrained scales often exceed the sigmoid co-domain.

Following the MLoRE setup (Yang et al., 2024), we train a ViT-S backbone using Adam with base learning rate $2 \times 10^{-5}$ and polynomial decay. Learning-rate multipliers of 100 and 10 are applied to TS$\sigma$BN and TSLN/TS$\sigma$LN layers respectively. Dropout and DropPath are disabled. Models are trained for 60k iterations.

### F.5 Swin Transformer

We follow the MTLoRA (Agiza et al., 2024) experimental protocol and use a pretrained Swin-T backbone. Intermediate representations from each stage are extracted and passed through a downsampling module before the task-specific HRNet heads. The original downsampler consists of a single 1×1 convolution per scale; in our variant, we insert TS$\sigma$BN modules both before and after this convolution. As in the ViT experiments, all remaining LayerNorm layers in the backbone are converted to TSLN.

In the PEFT setting most of the Swin-T backbone remains frozen, including the self-attention and MLP blocks. The patch embedding and patch merging layers stay trainable. When combining TS$\sigma$BN with LoRA, we add adapters only to the frozen linear layers and keep them shared across tasks, so they do not contribute task-specific capacity. We integrate our model using the MTLoRA codebase and keep all training hyperparameters unchanged. The only modification is the optimizer configuration required to support differential learning rates: as in all experiments we use a multiplier of 100 for TS$\sigma$BN parameters and 10 for LN.

## G   Additional Benchmarks

For completeness, we benchmark TS$\sigma$BN against multi-task optimization (MTO) methods. These approaches are orthogonal to architectural soft-sharing: instead of allocating task-specific parameters, they operate on a fully shared backbone and modify the optimization dynamics to mitigate conflict.

MTO methods fall into two major categories: loss-balancing strategies (e.g., UW, DWA, RLW) and gradient manipulation strategies (e.g., GradNorm, MGDA, PCGrad, CAGrad, Nash-MTL). Because they maintain a fully shared backbone, these methods are often highly parameter-efficient, but this comes with different trade-offs: they typically require additional computational overhead, involve external solvers to compute descent directions, and can significantly increase training time.

We evaluate TS$\sigma$BN within the LibMTL framework, using the NYUv2 setup with a pretrained DeepLabV3–ResNet50 backbone. We compare against the suite of optimization-based methods available in LibMTL, including UW (Kendall et al., 2017), GradNorm (Chen et al., 2018), MGDA (Sener & Koltun, 2018), DWA (Liu et al., 2019), GLS (Chennupati et al., 2019), PCGrad (Yu et al., 2020), GradDrop (Chen et al., 2020), IMTL (Liu et al., 2021b), GradVac (Wang et al., 2021), CAGrad (Liu et al., 2021a), Nash-MTL (Navon et al., 2022), and RLW (Lin et al., 2021).

Table 6 reports the results. TS$\sigma$BN achieves the highest overall multi-task gain among all optimization-based methods on top of the same HPS architecture, while maintaining the simplicity of a purely architectural soft-sharing mechanism.

| Method | Seg↑ | Depth↓ | Normal↓ | $\Delta$m (%)↑ |
|---|---|---|---|---|
| HPS | 53.93 | 0.3825 | 23.57 | 0.00 |
| +GradNorm | 53.91 | 0.3842 | 23.17 | 0.41 |
| +UW | 54.29 | 0.3815 | 23.48 | 0.44 |
| +MGDA | 53.52 | 0.3852 | 22.74 | 0.69 |
| +DWA | 54.06 | 0.3820 | 23.70 | -0.06 |
| +GLS | 54.59 | 0.3785 | 22.71 | 1.97 |
| +PCGrad | 53.94 | 0.3804 | 23.52 | 0.26 |
| +GradDrop | 53.73 | 0.3837 | 23.54 | -0.19 |
| +IMTL | 53.63 | 0.3868 | 22.58 | 0.84 |
| +GradVac | 54.21 | 0.3859 | 23.58 | -0.14 |
| +CAGrad | 53.97 | 0.3885 | 22.47 | 1.06 |
| +Nash-MTL | 53.41 | 0.3867 | 22.57 | 0.73 |
| +RLW | 54.04 | 0.3827 | 23.07 | 0.76 |
| **TS$\sigma$BN** | 53.78 | **0.3735** | **22.30** | **2.48** |

Table 6: Comparison with optimization-based MTL methods on the LibMTL NYUv2 benchmark using a pretrained DeepLabV3–ResNet50 backbone. All optimization methods are applied on top of the same HPS architecture. TS$\sigma$BN achieves the highest overall multi-task improvement among all optimization-based approaches.

## H   LoRA Capacity Sweep

To isolate the effect of LoRA capacity from the effect of per-task specialization, we sweep two LoRA variants on Swin-T / PascalContext following the protocol of Table 4: *Shared LoRA*, where a single low-rank adapter is shared across all tasks, and *Per-Task LoRA*, where each task has its own adapter. Both variants apply LoRA to the Q and V projections, matching the LoRA baseline in Table 4. Ranks span $r \in \{4, 8, 16, 32, 64\}$.

Adding shared capacity alone does not outperform the single-task baseline regardless of rank, even when matching the the number of parameters of TS$\sigma$BN. Per-Task LoRA surpasses STL at $r = 32$ (4.21M) and matches TS$\sigma$BN's $\Delta m$ only at $r = 64$ (6.47M, roughly 2$\times$ the parameter budget).

| Method | r | Seg ↑ | Parts ↑ | Sal ↑ | Norm ↓ | $\Delta m$ (%) | #P (M) |
|---|---|---|---|---|---|---|---|
| Shared LoRA | 4 | 67.85 | 55.70 | 60.35 | 19.23 | $-4.84$ | 2.02 |
| Shared LoRA | 8 | 67.89 | 55.80 | 60.96 | 19.01 | $-4.22$ | 2.09 |
| Shared LoRA | 16 | 67.93 | 56.09 | 61.49 | 18.81 | $-3.60$ | 2.23 |
| Shared LoRA | 32 | 67.94 | 56.35 | 61.96 | 18.62 | $-3.04$ | 2.51 |
| Shared LoRA | 64 | 67.82 | 56.65 | 62.64 | 18.42 | $-2.41$ | 3.08 |
| Per-Task LoRA | 4 | 69.33 | 56.45 | 61.45 | 18.16 | $-2.05$ | 2.23 |
| Per-Task LoRA | 8 | 69.60 | 56.78 | 61.86 | 17.95 | $-1.36$ | 2.51 |
| Per-Task LoRA | 16 | 69.78 | 57.21 | 62.47 | 17.76 | $-0.62$ | 3.08 |
| Per-Task LoRA | 32 | 70.07 | 57.71 | 62.90 | 17.45 | $+0.30$ | 4.21 |
| Per-Task LoRA | 64 | 70.33 | 58.23 | 63.72 | 17.31 | $+1.13$ | 6.47 |
| **TS$\sigma$BN** | – | 69.38 | 57.46 | 63.74 | 17.00 | $+0.91$ | **3.08** |
| **TS$\sigma$BN**(r=16) | 16 | 70.00 | 58.01 | 63.89 | 16.85 | $+1.63$ | **4.25** |

Table 7: LoRA capacity sweep on Swin-T / PascalContext. Shared LoRA never reaches positive $\Delta m$. Per-Task LoRA matches TS$\sigma$BN's $\Delta m$ only at $r = 64$, at roughly $2\times$ the parameter budget. The TS$\sigma$BN reference rows are reproduced from Table 4.

This sweep clarifies what TS$\sigma$BN contributes relative to LoRA. Per-Task LoRA adds task specific capacity through new trainable weight matrices, scaling parameter count with both rank and task count. TS$\sigma$BN achieves comparable specialization through bounded modulation with negligible parameter growth per task. Adding shared capacity on top (TS$\sigma$BN(r=16)) further improves $\Delta m$ to +1.63 while remaining below the parameter budget Per-Task LoRA requires to match the base TS$\sigma$BN, indicating that the two mechanisms compose rather than substitute.

