# OpenReview forum: "Simplifying Multi-Task Architectures Through Task-Specific Normalization"
_TMLR — Accepted by TMLR_

### Review · Reviewer_V6ia · 2026-04-29

**Summary Of Contributions:**

This paper argues that much of the complexity in modern multi-task learning may be unnecessary, and that task-specific normalization alone can provide a strong and efficient alternative to heavier task-routing or adapter-based designs. The authors first show that simply replacing shared normalization with task-specific normalization (TSBN) is already a surprisingly competitive baseline. They then propose TSσBN, which uses sigmoid-gated task-specific normalization with discriminative learning rates to softly allocate capacity across tasks while keeping the feature extractor fully shared. The method is evaluated across multiple settings, including CNNs trained from scratch, pretrained CNNs, ViT-based dense prediction, and a PEFT-style Swin-T setup, on NYUv2, Cityscapes, CelebA, and PascalContext. Across these settings, the method is generally strong and very parameter-efficient, often matching or outperforming substantially more complex alternatives. A particularly appealing aspect of the paper is the analysis enabled by the learned gates: the authors use them to study capacity allocation, filter specialization, and task relationships, which adds an interpretability component that is often missing in MTL work. Overall, I see the main contribution as a compelling case that task-specific normalization is not merely an implementation detail, but can serve as a principled and lightweight mechanism for interference reduction and task-aware capacity modulation in multi-task learning.

**Key Strengths**
1. **Clear and well-motivated central idea**
The paper makes a simple argument that is easy to understand and surprisingly impactful: task-specific normalization may already solve much of what more complex multi-task architectures are trying to address.
2. **Strong simplicity-to-performance ratio**
 One of the most appealing aspects of the work is that it achieves competitive or better results without adding heavy routing, expert modules, or large task-specific branches.
3. **Good empirical breadth**
 The method is evaluated across several architectures, training settings, and datasets, which makes the paper feel more convincing than a narrowly scoped benchmark win.
4. **Parameter efficiency is a genuine strength**
 The paper does not just report accuracy gains; it also shows that the method scales very lightly in parameter count, which is practically important for multi-task settings.
5. **Useful inclusion of TSBN baseline**
 I liked that the paper includes the simpler task-specific BN baseline. It helps isolate what is gained from task-specific normalization itself versus the proposed sigmoid-gated extension.
6. **Interpretability angle adds value**
 The analysis of task relationships, capacity allocation, and filter specialization makes the submission more insightful than a standard empirical paper.
7. **Overall presentation is fairly clear**
 The paper communicates its main message well, and the figures and tables generally support the narrative effectively.

**Key Weaknesses**

1. Some claims are broader than the evidence fully supports.  The paper occasionally frames the findings in a way that feels slightly too strong, especially when suggesting that complex MTL architectures may be unnecessary in general.
2. Scope is somewhat limited to the studied setting.  The method is mainly validated in single-domain vision MTL setups, so it is not yet fully clear how broadly the conclusions transfer to other modalities or more heterogeneous task settings.
3. Novelty is more in reframing and simplification than in a major conceptual leap . The contribution is interesting and useful, but part of the message is that a simpler design works well, which may feel less substantial to readers looking for a more fundamentally new MTL mechanism.
4. Some improvements are modest depending on the benchmark . While the overall picture is positive, not every result is dramatically stronger, so the paper is most convincing on efficiency and design elegance rather than uniformly large performance gains.
5. Optimization choices seem important to the final outcome . The discriminative learning-rate strategy appears to play a meaningful role, and the paper could make this dependence a bit more central in the framing of the contribution.
6. Interpretability claims could be phrased a little more carefully . The analysis is interesting, but some parts feel more descriptive than fully validated, so I would prefer slightly more caution in how these observations are presented.

**Additional Comments:**

I enjoyed reading this paper. Its biggest strength, in my view, is that it makes a simple idea feel genuinely consequential. The inclusion of TSBN as a strong minimal baseline is especially valuable, because it grounds the paper and prevents the contribution from seeming overstated. I also liked that the paper combines performance, efficiency, and analysis rather than focusing only on benchmark wins. The overall message is clear and potentially influential: many multi-task architectures may be more complicated than necessary, and normalization deserves more attention as a core design axis rather than a supporting detail. My main recommendation is therefore not to change the method, but to sharpen the framing. If the authors present the work slightly more conservatively, with a clearer accounting of scope and limitations, I think the paper will come across as both stronger and more credible.

**Audience:**

Yes

**Audience Explanation:**

Yes. I think this paper would be of interest to several parts of the TMLR audience, especially researchers working on multi-task learning, parameter-efficient adaptation, dense prediction, and model interpretability. The appeal is not only that the method performs well, but that it advances a broader design lesson: a very small architectural modification can compete with much more elaborate task-specific mechanisms. That message is relevant to people designing efficient MTL systems as well as to researchers questioning whether recent architectural trends are overcomplicating the problem. The paper also connects classical normalization layers with modern questions around capacity allocation and task interference, which gives it conceptual interest beyond the immediate benchmark results. The interpretability angle further increases its relevance, since the learned gates provide a readable view into task relationships and specialization patterns rather than only reporting final metrics. Overall, this is the kind of paper that could reshape how some researchers think about the default design space for MTL.

**Broader Impact Concerns:**

I do not have major broader-impact concerns specific to this submission. The work is primarily a methodological contribution in multi-task learning and, on its face, does not introduce a new application area with obvious direct societal risk. If anything, the method’s emphasis on simplicity, parameter efficiency, and interpretability could have positive downstream implications by reducing model complexity and making MTL systems easier to analyze. The main caution I would note is a standard one: like other generic representation-learning methods, this approach could eventually be used in high-stakes perception or decision systems, and in such settings performance and interpretability claims should not be overgeneralized beyond the tested benchmarks. But I do not think the paper raises any special ethical concern that would require a substantial broader-impact statement beyond a brief acknowledgment of this general consideration.

**Claims And Evidence:**

Yes

**Claims Explanation:**

The empirical evidence is generally convincing. The paper evaluates the proposed method across several architectures and protocols rather than on a single narrow benchmark, which strengthens the case that the idea is not overly specialized. The comparisons include both standard soft-sharing baselines and more recent MoE/PEFT-style approaches, and the results consistently support the main message that TSσBN offers a strong accuracy-efficiency tradeoff. I also appreciated that the paper includes a simpler TSBN baseline, because it helps isolate how much of the gain comes from task-specific normalization itself versus the sigmoid reparameterization and learning-rate strategy. The analysis sections are also a strength: the gradient conflict plots, task clustering, and pruning-based specialization study all point in a coherent direction and support the interpretation that the method is reducing interference while allocating capacity in a structured way. The appendix further strengthens the submission by showing that TSσBN is more robust than TSBN to aggressive learning-rate multipliers and by comparing against optimization-based MTL methods as well. That said, some of the stronger statements in the paper, such as the suggestion that complex MTL architectures may be unnecessary or that normalization is sufficient for the main MTL challenges, should be phrased a bit more carefully. The experiments strongly support the proposed method as a competitive and elegant alternative, but they do not fully establish that more complex approaches are broadly unnecessary in all regimes. So overall I would answer yes, with the caveat that a few of the higher-level claims should be softened to better match the evidence.

**Requested Changes:**

1. **Critical:** The paper should moderate some of its strongest claims. In particular, statements suggesting that normalization layers are “sufficient” for the main MTL challenges, or that complex MTL architectures may be unnecessary, feel somewhat broader than what is directly established by the experiments. The current evidence supports that TSσBN is a strong, simple, and efficient alternative across the tested settings, but it does not yet justify a universal conclusion across task families, modalities, and training regimes.
2. **Critical:** The paper should discuss limitations more explicitly. The method is motivated for single-domain MTL where all tasks share the same input distribution, and this assumption matters to the overall argument. A clearer discussion of when the approach may not work as well would improve the paper. For example, it would help to state more directly how the method is expected to behave when tasks are weakly related, when there is substantial domain shift, or when the architecture contains few or no normalization layers in its main computation path.
3. **Important but not critical:** The role of the discriminative learning-rate multiplier should be highlighted more clearly in the main paper. The appendix shows that this design choice matters substantially, and that TSσBN benefits from high multipliers in a way TSBN does not. Since this optimization choice is part of what makes the method effective, the main text should more clearly emphasize that the contribution is not only the normalization parameterization itself but also the associated training strategy.
4. **Important but not critical:** Some gains are modest in certain settings, so it would strengthen the submission to report uncertainty more consistently in the main tables, for example by including standard deviations or confidence intervals everywhere practical. The paper states that results are averaged over three runs, which is good, but more visible reporting of variability would make the claims easier to assess.
5. **Important but not critical:** The interpretability section is interesting and promising, but some parts would benefit from a clearer distinction between descriptive analysis and validated causal explanation. For example, the task-relationship and filter-specialization analyses are intuitive and useful, but the paper should be a bit more careful about how strongly these analyses are interpreted. A short discussion of what these gate-based measures can and cannot tell us would improve this section.
6. **Suggestion:** Since the paper makes a broader argument about reevaluating complexity in MTL, it would help to include a short discussion comparing where TSσBN is likely to be preferable to optimization-based MTL methods versus where those methods might still be attractive, even though Appendix G already provides additional comparisons.

---

> ### Author Response · Authors · 2026-05-11
>
> We thank the reviewer for a thoughtful and constructive review. We are grateful for both the recognition of simplicity as a contribution and the specific guidance on framing. We agree with the central recommendations and address each requested change below.
>
> **Critical 1 (moderating strong claims).** We have revised the framing throughout the abstract, contributions list, and conclusion to scope claims to the evaluated settings rather than implying universal conclusions.
>
> **Critical 2 (limitations).** We have added an explicit limitations and future-directions paragraph to the conclusion, covering the bounded $\sigma$-gate (which keeps tasks on a comparable scale but prevents amplification beyond a filter's native magnitude), the descriptive-only nature of the importance-matrix analysis (with an explicit dissimilarity regularizer flagged as a natural extension for highly related tasks), and the deliberate single-domain scope with references to the multi-domain conditional BN literature.
>
> **Important 1 (discriminative learning rate in framing).** We have elevated discriminative learning rates such that the abstract now describes TS$\sigma$BN as combining bounded $\sigma$-gated normalization with discriminative learning rates; the corresponding contributions bullet has been rewritten; and §4 has been restructured so the training strategy is presented immediately after the parameterization, before the analysis paragraphs.
>
> **Important 2 (uncertainty reporting).** We have added standard deviations (mean$_{(\text{std})}$ over three random seeds) to the main from-scratch results table for all methods, and to the LibMTL pretrained-backbone table for our re-runs (Cityscapes, all methods). For the remaining tables and baselines drawn from prior publications, values are reported as published as rerunning all baselines is not feasible.
>
> **Important 3 (interpretability disclaimers).** We have added a short paragraph at the end of §5 distinguishing descriptive observations from causal claims. We also frame the analysis positively as a diagnostic tool, useful for inspecting MTL dynamics when task relationships are not known a priori.
>
> **Suggestion (TS$\sigma$BN vs optimization-based MTL).** We have added a short discussion paragraph in the conclusion contrasting the two families along orthogonal axes: optimization-based methods scale naturally with task count but rely on external solvers that become prohibitive on larger models, whereas architectural methods like TS$\sigma$BN scale well with model size but are bounded by what their modulation mechanism can express. The two families are framed as complementary rather than competing, with the empirical comparison in Appendix G referenced.
>
> We thank the reviewer once more for a review that has genuinely improved the paper.

---

### Review · Reviewer_mNxD · 2026-05-04

**Summary Of Contributions:**

Summary

This paper studies multi-task learning (MTL) and argues that complex architectural designs may be unnecessary. The authors show that replacing shared normalization layers with task-specific variants already provides competitive performance. Based on this observation, they propose Task-Specific Sigmoid Batch Normalization (TSσBN), a lightweight mechanism that allows tasks to softly allocate network capacity while fully sharing feature extractors. The method is evaluated on several benchmarks and is shown to achieve strong performance while remaining parameter-efficient.

Strengths

1. Multi-task learning (MTL) is an important problem.
2. The introduction and related work are comprehensive.

Weaknesses

1. The method is mainly developed for convolutional layers with batch normalization, while modern architectures more commonly rely on other normalization techniques (e.g., layer normalization in Transformer blocks).
2. The method is relatively straightforward, as it essentially introduces task-specific trainable rescaling factors.

**Audience:**

Yes

**Audience Explanation:**

The paper is about multi-task learning, which is an important topic.

**Claims And Evidence:**

Yes

**Claims Explanation:**

Numerical experiments are provided to show the performance of the proposed architecture.

**Requested Changes:**

1. For the batch normalization in each convolutional layer, is $\gamma_t$ the only trainable parameter? Is $\gamma_t$ a scalar or a vector?
2. What specific training algorithm is used in this paper? There are multiple MTL training paradigms (e.g., transfer learning, meta-learning). It would be helpful to include explicit training pseudocode.
3. In the second paragraph of the conclusion, the authors claim that the proposed method matches or outperforms state-of-the-art MoE-style and PEFT-based MTL methods. This seems exaggerated, as MoE and PEFT are more general approaches and are not restricted to CNNs with batch normalization.
4. The paper focuses on single-domain MTL where all tasks share the same input distribution. What happens when the input distributions differ? Would the performance degrade significantly?

---

> ### Author Response · Authors · 2026-05-11
>
> We thank the reviewer for the assessment and feedback, and address each point below.
>
> **W1 / RC3 (Applicability to LayerNorm/Transformers).** TS$\sigma$BN is evaluated on Transformer backbones as well as CNNs. Tables 3 and 4 report results on ViT-S (PascalContext 5-task, MLoRE protocol) and Swin-T (PascalContext 4-task, MTLoRA protocol), using the LayerNorm variant described in §4 and Appendix F.3. The MoE and PEFT comparisons are both conducted on Transformers. The claim about matching or outperforming MoE-style and PEFT-based MTL methods therefore refers to comparisons on the same backbones these methods were designed for. We have added a scope qualifier in the conclusion to make it more visible.
>
> **W2 (Simple rescaling factors).** The implementation is intentionally simple, and we view this as a feature rather than a limitation; the contribution goes beyond per-task scaling parameters. Our design adds (i) $\sigma$-gating, which bounds the scaler and stabilizes training under aggressive learning rates; (ii) discriminative learning rates on $\sigma$BN parameters, which allow tasks to specialize before the shared backbone shifts; and (iii) the extraction of an interpretable filter-importance matrix used in Sections 5 and 7. These choices are what allow the method to retain a simple form while still functioning as an effective soft-sharing mechanism. In the revised version, we have made discriminative learning rates more prominent in the abstract, contributions list, and §4.
>
> **RC1 ($\gamma$ scalar or vector?).** $\gamma_t$ is a per-channel vector with dimension equal to the number of feature channels at that layer ($\gamma_t \in \mathbb{R}^C$ for a Conv–BN block with $C$ output channels), and $\sigma$ is applied elementwise. This per-channel structure is what enables the task-filter importance matrix used in our analysis. We have tightened the text immediately preceding Equation 3 to state this explicitly.
>
> **RC2 (Training paradigm and pseudocode).** We use the standard joint multi-task training formulation: summed task losses and shared-backbone updates, with the $\sigma$BN parameters receiving a higher learning rate via a fixed multiplier. This is now made explicit in a new algorithm block added at the start of Appendix F.
>
> **RC4 (Different input distributions across tasks).** The single-domain scope is deliberate. Multi-domain MTL is where conditional BN was developed to alleviate mismatched statistics across domains, and is a well-studied setting (§2). Our contribution is showing the mechanism is suprisingly useful in single-domain MTL, where no such statistical mismatch motivates it. We expect TS$\sigma$BN to remain effective in multi-domain settings, but demonstrating it there would overlap with prior work rather than isolate the single-domain finding. We have added this scope clarification to the limitations paragraph in the conclusion.

---

### Review · Reviewer_2wmR · 2026-05-12

**Summary Of Contributions:**

###  Summary

The paper focuses on multi-task learning (MTL), where a single model is trained to solve multiple vision tasks jointly. It proposes TS$\sigma$BN, a simple task-specific normalization approach that keeps the main vision backbone shared while using sigmoid-gated normalization parameters to softly allocate network capacity across tasks. This provides a parameter-efficient alternative to more complex MTL architectures that rely on task-specific modules, routing, or experts. The paper reports positive results on several standard MTL benchmarks, including NYUv2, Cityscapes, and CelebA, showing that TS$\sigma$BN is competitive with or improves over a range of existing baselines while adding relatively few task-specific parameters.

### Strengths

- The paper proposes a lightweight alternative to complex MTL architectures by using task-specific normalization layers with minimal parameter overhead.

- The method is evaluated across multiple datasets, architectures, and settings, including CNNs, transformers, from-scratch training, and pretrained backbones.

### Weaknesses

The paper’s motivation should be clarified and better positioned within the current literature. It is unclear if the issue persists in more recent, well-trained vision backbones such as DinoV3 [a] or even older backbones such as CLIP [b]. If the problem does not persist in those models, the work should motivate use cases in which this proposed framework could be useful.

Novelty: The proposed method appears to have limited methodological novelty, as it primarily extends existing sigmoid BatchNorm [c] by introducing task-specific copies for each task in a multi-task network. While the empirical findings may be useful, the core architectural change seems incremental and may not meet the TMLR bar for modest novelty.

Task specialization: In Section 5.2, the paper computes cosine similarities between task-specific filter-importance vectors learned by TS$\sigma$BN and shows that the resulting task clusters are semantically meaningful. However, it is unclear whether this clustering is unique to the proposed method or whether similar task relationships would emerge in existing MTL baselines. The analysis would be stronger if the authors compared against baselines without task-specific normalization and showed whether those methods fail to produce similarly meaningful or stable task clusters.

Experiments: Several experiments rely on standard but relatively older backbones and well-established MTL benchmarks. The paper would be stronger if it included results with more recent foundation-style vision backbones, such as DINOv3 [a] or CLIP [b], to demonstrate that the proposed approach remains effective in more modern and less saturated settings.

LoRA results: The proposed method outperforms the reported LoRA baseline in the PEFT experiments. To better contextualize this result, it would be useful to include additional LoRA baselines with varying ranks and report performance as a function of trainable parameter count. This would clarify whether the observed gap persists as LoRA capacity increases and how TS$\sigma$BN compares across a broader range of parameter-efficiency trade-offs.

Minor: How do you pronounce the name of the method?

References:

[a] DINOv3. arXiv 25.

[b] Learning Transferable Visual Models From Natural Language Supervision. ICML 21.

[c] Receding Neuron Importances for Structured Pruning. arXiv 22.

**Audience:**

No

**Audience Explanation:**

In my opinion, the paper focuses on a relatively mature problem, and its relevance to the current research landscape is not fully clear. As a result, it is uncertain whether the findings will be of broad interest to the TMLR audience. I have asked the reviewers to provide additional details and context, which may change my assessment of the paper.

**Claims And Evidence:**

Yes

**Claims Explanation:**

The claims are mostly supported in the paper. I have requested additional experiments.

**Requested Changes:**

I've listed the requested changes in the weaknesses section of the review. I'm relisting them here again:

- Clarify the paper’s motivation and position it in terms of recent work.
- Show whether the identified problem persists in modern, well-trained backbones, or clearly motivate use cases where the framework remains useful.
- Add evidence that the proposed task-specialization clusters are unique to TS$\sigma$BN by comparing them with clusters from existing MTL baselines.
- Expand the experiments to include more recent foundation-style vision backbones such as DINOv3 or CLIP.
- Show whether the performance gap over LoRA persists as LoRA capacity increases.

---

> ### Author Response · Authors · 2026-05-21
>
> We thank the reviewer for the thoughtful review. We address each point below.
>
>
> ### 1. Relevance
>
> *TLDR: Foundation models have changed how models are trained, but not the structural problems that arise when multiple tasks share trainable parameters. Task interference and dominance depend on the relationships between tasks, not on backbone quality. Tables 3 and 4 evaluate this regime, using contemporary transformers (ViT, Swin).*
>
> We agree that foundation backbones have substantially improved the representations available for downstream tasks and the quality of single-task fine-tuning. However, multi-task fine-tuning is a different setting: it requires joint optimization of shared trainable parameters, and wherever that occurs, interference and capacity allocation reappear. These are properties of the joint loss over related tasks, not of the backbone. Foundation models have moved the scope of MTL from the entire network to the trainable subset, not dissolved it. If they had, recent transformer-specific MTL methods such as MTLoRA (CVPR 2024) and MLoRE (CVPR 2024) would not exist, and would not report meaningful gaps over single-task or hard-sharing baselines. Our evaluation shows that we match or outperform these state-of-the-art approaches in this regime.
>
> Furthermore, we would like to note that from-scratch training still remains a relevant paradigm to this day, especially for resource-constrained deployment and in applications where pretrained backbones underperform due to domain shift.
>
> We appreciate the suggestion and agree that this contextualization improves the positioning of our paper. We have sharpened the introduction to make this more visible.
>
> ### 2. Modern backbones
>
> We considered evaluating on CLIP or DINOv3 but view this as outside the scope of a revision. CLIP (ICML 2021) predates the pretrained ViT and Swin backbones we already evaluate on, so it would not address the request for a more modern setting. DINOv3 is recent enough that no established multi-task evaluation protocol exists to our knowledge. A fair comparison would require defining a new MTL protocol and porting every baseline in Tables 3 and 4 to that backbone, which is a separate contribution rather than a revision item. The findings of our paper (σ-normalization, the importance matrix, discriminative learning rates) are properties of the mechanism and transfer across backbones as evidenced by consistent gains on SegNet, ResNet, ViT-S, and Swin-T.
>
> ### 3. LoRA capacity sweep
>
> We thank the reviewer for the suggestion to isolate the effect of extra capacity and specialization. To emphasize the tradeoff we test two variants: **Shared LoRA**, where a single low-rank adapter is shared across tasks, and **Per-Task LoRA**, where each task has its own adapter. Ranks span r ∈ {4, 8, 16, 32, 64}. The implementation is in line with the baseline in Table 4. We focus on performance vs parameter count.
>
> | Method | r | Seg ↑ | Parts ↑ | Sal ↑ | Norm ↓ | Δm % | #P (M) |
> |---|---|---|---|---|---|---|---|
> | Shared LoRA | 4 | 67.85 | 55.70 | 60.35 | 19.23 | −4.84 | 2.02 |
> | Shared LoRA | 8 | 67.89 | 55.80 | 60.96 | 19.01 | −4.22 | 2.09 |
> | Shared LoRA | 16 | 67.93 | 56.09 | 61.49 | 18.81 | −3.60 | 2.23 |
> | Shared LoRA | 32 | 67.94 | 56.35 | 61.96 | 18.62 | −3.04 | 2.51 |
> | Shared LoRA | 64 | 67.82 | 56.65 | 62.64 | 18.42 | −2.41 | 3.08 |
> | Per-Task LoRA | 4 | 69.33 | 56.45 | 61.45 | 18.16 | −2.05 | 2.23 |
> | Per-Task LoRA | 8 | 69.60 | 56.78 | 61.86 | 17.95 | −1.36 | 2.51 |
> | Per-Task LoRA | 16 | 69.78 | 57.21 | 62.47 | 17.76 | −0.62 | 3.08 |
> | Per-Task LoRA | 32 | 70.07 | 57.71 | 62.90 | 17.45 | +0.30 | 4.21 |
> | Per-Task LoRA | 64 | 70.33 | 58.23 | 63.72 | 17.31 | +1.13 | 6.47 |
> | **TSσBN** | – | **69.38** | **57.46** | **63.74** | **17.00** | **+0.91** | **3.08** |
> | **TSσBN(r=16)** | 16 | **70.00** | **58.01** | **63.89** | **16.85** | **+1.63** | **4.25** |
>
> We make two findings:
> - Shared LoRA never reaches positive relative performance, even at r=64 with the same parameter budget as TSσBN (3.08M). Capacity alone, without per-task specialization or modulation, is a inefficient way to uncrease performance.
> - Per-Task LoRA reaches positive performance only at r=32 and matches pure TSσBN's Δm at r=64 with ~2× parameters. While task-specific capacity has a better rate of improvement it is less efficient than modulation.
>
> This sweep also clarifies what TSσBN contributes. While adding per-task capacity improves performance, it has to scale parameter count with rank and number of tasks. TSσBN achieves comparable performance, with negligible parameter growth per task, and any additional shared capacity (TSσBN(r=16)) is used more efficienty.
>
> We add these findings to the revision in section 6.1 as well as the appendix.

---

> ### Author Response · Authors · 2026-05-21
>
> ### 4. Task-relationship comparison
>
> We thank the reviewer for this suggestion. We would like to emphasize that existing MTL baselines do not produce task relationships natively. In order to enable a comparison, we construct an analogous importance matrix for a hard-parameter-sharing baseline by ablating each filter individually and measuring the resulting per-task performance drop on the validation set. This produces a task-filter importance matrix, but requires a full evaluation pass per filter - unlike our method, which extracts it from trained parameters at zero additional cost. For consistency, we use the same clustering pipeline as in section 5.2: cosine similarity between task vectors, hierarchical clustering, and co-occurrence aggregation across seven seeds.
>
> We make two findings:
> - The core of the clusters is consistent across methods. The strongest task pairs the pruning baseline recovers - (Bags_Under_Eyes, Big_Nose), (Heavy_Makeup, Wearing_Lipstick), (High_Cheekbones, Smiling), (Chubby, Double_Chin) - appear as strict subsets of TSσBN's clusters. This shows that the dominant structure reflects real shared filter usage and is not an artifact of our method.
> - The broader cluster structure and its stability are not similar. TSσBN recovers semantically coherent groups, with a mean Spearman rank correlation of **0.80**. The pruning baseline reaches only **0.27** and at that level only the strongest pairs survive aggregation while the larger hierarchy is noisy.
>
> We added this experiment to the discussion in 5.2 as well as the appendix.

---

### Decision · Action_Editor_6XVZ · 2026-06-19

**Recommendation:** Accept as is

**Audience:**

Yes

**Audience Explanation:**

In a research landscape increasingly dominated by complex multi-task architectures like MoEs or dense adapters, this work offers a valuable minimalist alternative. It demonstrates that task-specific normalization alone can deliver competitive performance at a fraction of the parameter cost. Furthermore, its ability to extract an interpretable task-filter importance matrix at zero additional computational cost provides a practical diagnostic framework that will interest the multi-task learning and PEFT communities.

**Claims And Evidence:**

Yes

**Claims Explanation:**

The submission are supported by thorough empirical evidence across multiple datasets and architectures.